# MULTI-FEATURE QUANTIZED SELF-ATTENTION FOR FAIR LARGE LANGUAGE MODELS

**Jaeil Park & Sung-Bae Cho**
Department of Computer Science
Yonsei University
{wodlf603,sbcho}@yonsei.ac.kr

## ABSTRACT

Large language models (LLMs) often encode social biases tied to sensitive features such as race and gender, undermining fairness in downstream tasks even after instruction tuning. Conventional debiasing methods require expensive fine-tuning, are tied to specific architectures, or operate only at the input or decoding stage while neglecting attention-level representations, which can result in compromised task performance. Moreover, most approaches are tailored to single-attribute settings and do not explicitly address scenarios with multiple, overlapping protected attributes and their intersections. This paper proposes a novel method of multi-feature quantized attention regularization (MQAR) to mitigate multi-feature bias by injecting a structured quantization into frozen self-attention layers. MQAR disentangles attribute-specific activations through vector-quantized regularization and uses a discriminator-guided autoencoding regularizer to adversarially suppress protected-attribute information while preserving task-relevant semantics. Crucially, the proposed method operates without modifying the backbone parameters or accessing pre-training data, ensuring architecture-agnostic applicability and minimizing representation distortion. MQAR is evaluated on five diverse LLMs (BERT, T5, GPT-Neo, Mixtral, and LLaMA 3.2) using three standard bias benchmarks (WinoBias, StereoSet, and CrowS-Pairs). Across these models, MQAR consistently reduces bias for multiple protected attributes and their intersections while maintaining downstream accuracy within at most 0.4 %, on average, of non-debiased baselines on sentiment analysis, abusive language detection, and text generation tasks. These findings highlight quantized attention regularization as a scalable and effective method for mitigating social bias in modern language models.

## 1 INTRODUCTION

Large language models (LLMs) have shown remarkable performance across a wide range of natural language processing (NLP) tasks. By leveraging self-attention mechanisms, LLMs encode rich contextualized representations that can generalize well to downstream applications such as sentiment analysis, natural language inference, and text generation (Sun et al., 2019; Alaparthi & Mishra, 2021; Lin & Su, 2021; Zhu et al., 2020). However, it has been widely observed that these models also absorb and propagate social biases present in the training data, including those related to gender, race, and religion (Schramowski et al., 2022). Such biases not only undermine fairness but also introduce systemic risks when LLMs are deployed in real-world applications.

Recent work has shown that bias in LLMs is not solely inherited from data but is often amplified by the internal self-attention structures of the models (Jiang et al., 2022). The attention mechanism entangles sensitive feature information with semantic content, causing the model to produce biased outputs even in seemingly neutral contexts (Liu et al., 2024). For example, in abusive language detection, sentences containing female-related terms are more likely to be misclassified as offensive, reflecting a learned association between gender and abuse (Park et al., 2018; Park & Cho, 2025). These representational biases persist even when downstream classifiers are trained separately, as the bias is embedded within the latent representations themselves, and they remain measurable even in instruction-tuned and alignment-optimized LLMs.

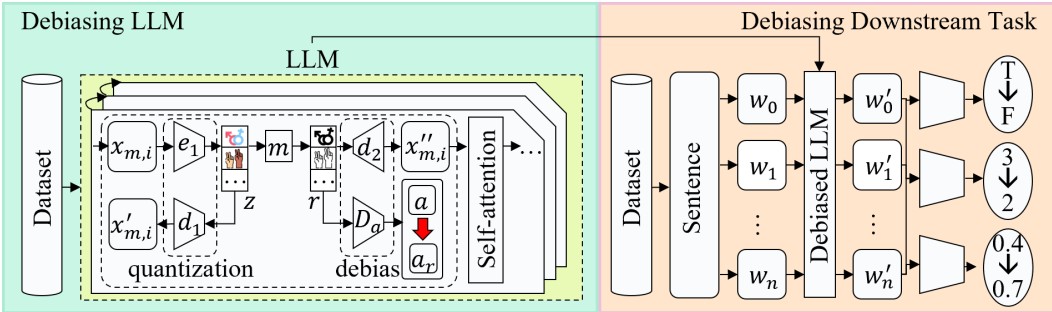

Figure 1: Overview of the proposed MQAR method. Token sequences are first encoded via the frozen self-attention layers of a pre-trained LLM. Structured quantization is applied to the attention outputs to isolate attribute-specific activations, followed by an adversarial autoencoder that regularizes the latent space to remove biased information without compromising semantic consistency.

To address this, several debiasing methods have been proposed, including dataset reweighting (Mozafari et al., 2020), input perturbation (Manerba & Tonelli, 2021), and fairness-aware fine-tuning (Yu et al., 2023). More recently, inference-time techniques based on prompting and decoding have also been explored to steer model outputs without modifying model parameters. However, these approaches suffer from two key limitations. First, they often rely on retraining or fine-tuning the backbone of the LLM, which is computationally expensive and often architecture-dependent. Second, many methods are designed for a single protected attribute and do not scale well to real-world scenarios involving multiple, overlapping protected attributes and their intersections (Zhao et al., 2023). Consequently, there remains a need for a model-agnostic, lightweight debiasing method that directly targets representation-level bias in multi-attribute settings without modifying the core parameters of the LLM.

To overcome this problem, this paper proposes Multi-feature Quantized Attention Regularization (MQAR), a novel method for mitigating social bias in LLMs by intervening in their frozen self-attention layers. As shown in Figure 1, MQAR introduces a structured quantization process into attention outputs to disentangle attribute-specific activations, followed by an adversarial autoencoder that regularizes the representations to suppress bias while preserving semantic information. In contrast to prior approaches, MQAR operates directly on the internal attention representations of a frozen backbone and is trained with multi-feature supervision so that a single MQAR module can jointly debias multiple protected attributes (e.g., gender and race) without access to the original training data. Moreover, MQAR leaves the LLM backbone untouched, enabling generalized deployment across diverse pre-trained models, including BERT, T5, GPT-Neo, Mixtral, and LLaMA 3.2.

MQAR is evaluated on two widely used bias benchmarks, WInoBias, StereoSet, and CrowS-Pairs. Across five LLMs, MQAR consistently reduces bias for multiple protected attributes and their intersections, while maintaining downstream accuracy within at most 0.4% of non-debiased baselines on tasks such as sentiment analysis, abusive language detection, and text generation. These results position MQAR as a scalable, architecture-agnostic, and fine-tuning-free debiasing solution. Its ability to mitigate multi-attribute bias without retraining makes it especially practical for real-world applications where retraining is infeasible.

## 2 RELATED WORKS

### 2.1 LARGE LANGUAGE MODELS AND THEIR BIAS

Transformer-based LLMs such as BERT (Devlin et al., 2018), T5 (Raffel et al., 2020), GPT-3 (Brown et al., 2020), Mistral (Jiang et al., 2023), and LLaMA (Touvron et al., 2023) have achieved remarkable performance on a wide range of NLP tasks. These models learn rich contextual representations through self-attention over large-scale corpora, which inherently reflect and reproduce social biases present in the data (Schramowski et al., 2022). Such biases have been shown to propagate to

downstream applications, leading to discriminatory behavior in practical tasks such as abusive language detection, sentiment analysis, and coreference resolution, and they remain measurable even in instruction-tuned or alignment-optimized variants.

To evaluate these effects, several benchmark datasets have been developed. SEAT (May et al., 2019) measures sentence-level association bias, WinoBias (Zhu et al., 2020) focuses on gendered coreference resolution, StereoSet (Nadeem et al., 2020) and CrowS-Pairs (Nangia et al., 2020) target stereotypical associations across domains, BiasBios (De-Arteaga et al., 2019) evaluates occupational bias using gender-masked bios, BBQ (Parrish et al., 2022) explicitly probes social and intersectional bias in question–answering settings, and CoBia (Nikeghbal et al., 2025) constructs lightweight adversarial conversations in which an LLM first produces a biased claim. These datasets collectively provide a comprehensive suite for evaluating representational and behavioral biases in LLMs under both static and interactive settings. This paper primarily employs WinoBias, StereoSet, and CrowS-Pairs to assess how well MQAR mitigates biases across multiple protected attributes.

## 2.2 Bias Mitigation in Language Models

Bias mitigation strategies for language models are typically categorized into embedding-level, model-level, and inference-time approaches. Embedding-based methods, such as SentDebias (Liang et al., 2020), INLP (Ravfogel et al., 2020), and OSCaR (Dev et al., 2020), aim to remove protected-attribute information via projection (e.g., removing directions aligned with protected attributes) or representation alignment. However, these methods may compromise the expressiveness of learned features by overly constraining the latent space, thus reducing its capacity to encode task-relevant information (Shin et al., 2020; Kaneko & Bollegala, 2019).

Model-level approaches, including ADELE (Lauscher et al., 2021) and FaRM (Chowdhury & Chaturvedi, 2022), inject fairness objectives into training loss functions or introduce debiasing modules during training. While effective, such methods often require access to training data and extensive fine-tuning of the backbone, which limits their scalability and adaptability to new domains and proprietary LLMs.

Inference-time approaches aim to mitigate bias during model inference by manipulating prompts or controlling decoding strategies (Fatemi et al., 2021), and more recent methods such as CRISPR, CPAD, and RB (e.g., Yang et al., 2024; Dai et al., 2024; Kim et al., 2025) further refine outputs by adjusting generation trajectories without changing model parameters. These techniques are attractive because they are model-agnostic and parameter-free, but they do not directly modify internal representations, and therefore may leave attention-level biases intact.

Despite recent progress, most prior work targets a single protected attribute at a time and provides limited support for scenarios involving multiple, intersecting protected features (e.g., gender and race) (Zhao et al., 2023). Moreover, existing methods are typically tied either to embedding-level projections, full-model fine-tuning, or purely output-level inference-time control. In contrast, MQAR directly targets internal attention activations in frozen LLMs by introducing a quantized bottleneck and an adversarial autoencoder that jointly debias multiple protected features with a single plug-in module, without access to pre-training data or modification of backbone parameters. To the best of our knowledge, MQAR is among the first attention-level debiasing methods that operate entirely on frozen LLMs while explicitly addressing multiple protected attributes and their intersections.

## 3 Quantized Regularization of Self-Attention

### 3.1 Representing Self-Attention of LLMs

To formalize the intervention point for the proposed regularization, this section begins by briefly reviewing the self-attention mechanism in transformer-based LLMs. Given an input token sequence $W_t = \{w_0, \ldots, w_n\}$, a pre-trained LLM encodes it through a stack of multi-head self-attention layers, producing contextualized representations at each layer. Let $X_{i-1} \in \mathbb{R}^{n \times d}$ denote the hidden state at layer $i - 1$. The attention operation at layer $i$ is defined as:

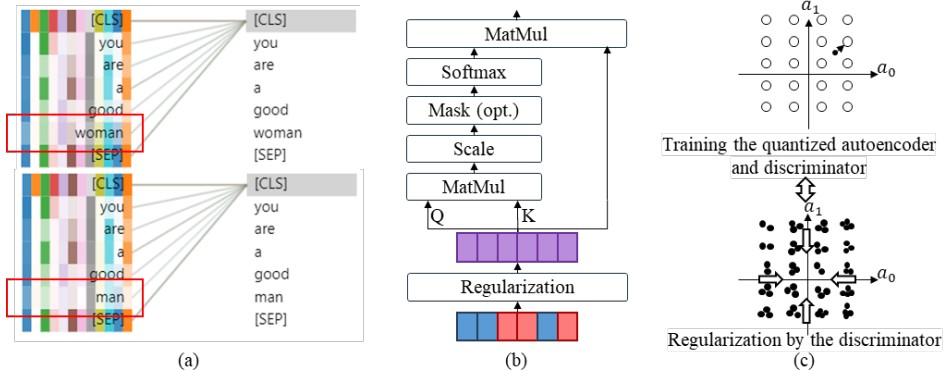

Figure 2: (a) Visualization of attention heatmaps in BERT for gender-swapped input pairs, revealing differential focus induced by gender bias. (b) Schematic overview of MQAR's integration into the self-attention layer via pre-attention regularization. (c) Conceptual diagram illustrating how quantized regularization compresses and disentangles protected attribute signals from semantic representations.

$$X_i = \text{softmax}\left(\frac{Q_i K_i^T}{\sqrt{d_k}}\right) V_i, \quad Q_i = X_{i-1} W_i^Q, \quad K_i = X_{i-1} W_i^K, \quad V_i = X_{i-1} W_i^V \quad (1)$$

where $d_k$ denotes the dimension of the key vectors. While this attention mechanism effectively captures contextual dependencies, it can also entangle task-irrelevant signals such as sensitive feature information. As illustrated in Figure 2(a), gender-swapped sentence pairs elicit markedly different attention maps, indicating that the attention mechanism encodes attribute-specific signals.

To mitigate this, as illustrated in Figure 2(b, c), this paper introduces a layer-wise quantized regularization (QR) module that transforms the input $X_{i-1}$ into a debiased representation $\tilde{X}_{i-1}$:

$$\tilde{X}_{i-1} = QR_{i-1}(X_{i-1}) \quad (2)$$

$$X_i = \text{softmax}\left(\frac{\tilde{X}_{i-1} W_i^Q (\tilde{X}_{i-1} W_i^K)^T}{\sqrt{d_k}}\right) \tilde{X}_{i-1} W_i^V \quad (3)$$

Section 3.2 describes a formal explanation of the regularization. This design enables MQAR to suppress sensitive feature information within the attention computation itself, improving fairness without degrading contextual expressiveness. Crucially, the method requires no modification to LLM parameters, making it applicable to frozen pre-trained models.

### 3.2 REGULARIZING SELF-ATTENTION WITH QUANTIZED ADVERSARIAL AUTOENCODER

The proposed method mitigates representational bias in self-attention by applying structured regularization over latent representations within a quantized adversarial autoencoder. Autoencoders compress high-dimensional embeddings into a lower-dimensional latent space, and when combined with adversarial learning, allow suppression of sensitive attribute signals while retaining semantic content.

**Structure.** Formally, given a token representation $x_{(l,i)}$ at layer $l$ and position $i$, an encoder $e_1$ first maps it to a latent vector $z_{(l,i)} = e_1(x_{(l,i)})$. A reconstruction decoder $d_1$ then produces $x'_{(l,i)} = d_1(z_{(l,i)})$, and a reconstruction loss encourages $x'_{(l,i)}$ to remain close to $x_{(l,i)}$, ensuring that $z_{(l,i)}$ preserves the original semantic content. To introduce a debiasing bottleneck, MQAR applies a structured vector quantizer $Q$ to obtain a mutated latent representation $r_{(l,i)} = Q(z_{(l,i)})$, which

is fed forward to subsequent layers in place of the original representation. A second decoder $d_2$ reconstructs the token representation from this quantized latent $x''_{(l,i)} = d_2\big(r_{(l,i)}\big)$, and an additional reconstruction term encourages $x''_{(l,i)} \approx x_{(l,i)}$ so that quantization and debiasing do not excessively distort task-relevant semantics.

For debiasing, each token (or sentence) is annotated with a multi-hot protected-attribute label vector $\mathbf{a} \in \{0,1\}^M$, where $M$ is the number of protected attributes (e.g., gender, race, religion) and each dimension indicates the presence of a particular attribute. A discriminator $D_a$ takes the mutated latent $r_{(l,i)}$ as input and predicts $\hat{\mathbf{a}} = D_a(r_{(l,i)})$. During training, $D_a$ is optimized to correctly classify $\mathbf{a}$, while the encoder–quantizer pair $(e_1, Q)$ is trained adversarially to obfuscate $\mathbf{a}$. This multi-hot supervision discourages the model from encoding correlated attribute signals in a shared latent subspace and drives $r_{(l,i)}$ toward a representation that retains semantic information but suppresses multiple protected features jointly.

Accordingly, the quantized regularization (QR) module at layer $l$ and position $i$ can be viewed as the composition $QR_i(\cdot) = d_2\big(Q(e_1(\cdot))\big)$, which maps an attention output to its debiased, reconstructed counterpart before it is passed to the remaining layers.

**Training 1: Training quantized autoencoder and discriminator.** According to Hsu et al. (2024), the proposed method uses two types of loss to train a quantized autoencoder, consisting of encoder $e_1$ and decoder $d_1$, in a differentiable manner. The two losses enable the autoencoder to learn the discrete representation $\hat{a} \in \mathbb{Z}_K^{m'}$ of $a$ in the latent space through the data and to guide the latent vector $z$ of the input data towards $\hat{a}$ with some integer $K$ and $m'$. Here, $\hat{a} \in \mathbb{Z}_K^{m'}$ denotes the quantized latent vector derived from a codebook of size $K$, representing discrete prototypes aligned with protected attributes. The encoder $e_1$ maps inputs to a continuous latent $z$, which is then quantized into $\hat{a}$ using nearest-neighbor search. The straight-through gradient estimator enables the propagation of gradients through the non-differentiable quantization step. The quantization loss $L_{quantize}$ pulls the discrete values constituting $z$ unilaterally towards the encoder's pre-quantized continuous output. This adjustment is necessary to optimize $\hat{a}$, because the straight-through gradient estimation decouples $\hat{a}$ from the computation graph. In contrast, the commitment loss $L_{commit}$ ensures that the pre-quantized representation, which receives gradients from downstream computations, remains close to the codebook entries. Although this issue is significant in vector quantization, using scalars instead of high-dimensional vectors mitigates the problem. This allows for a substantial reduction in the quantization and commitment losses while maintaining training stability, providing the model with essential flexibility to reorganize the discrete latent space.

$$L_{quantize} = \|StopGradient(z) - \hat{a}\|_2^2 \tag{4}$$

$$L_{commit} = \|z - StopGradient(\hat{a})\|_2^2 \tag{5}$$

Training of the autoencoder, decoder $d_1, d_2$, and discriminator occurs simultaneously. The losses for the decoder and discriminator, which are composed of MLPs, use BCE (Binary Cross-Entropy) to compare with $x$ and $a$, respectively. Equation (6) represents the overall loss term at this stage.

$$L = L_{quantize} + L_{commit} + L_{d_1} + L_{d_2} + L_{D_a} \tag{6}$$

**Training 2: Regularization by discriminator.** Subsequently, the discriminator $D_a$ is fixed, and protected feature labels of the mutated latent vector $a = \{a_0, ..., a_m\}$ are randomized as $a^r = \{a_0^r, ..., a_m^r\}$. As $z$ and $r$ are regularized, while $D_a$ maintains to discriminate the protected feature label of them appropriately, $z$ and $r$ are learned not to be distinguished by $D_a$. By iterating learning the discriminator and randomizing the label, $z$ and $r$ are learned to disentangle for protected features (Makhzani et al., 2015; Park & Cho, 2023).

The goal is to optimize the latent representation $r$ such that it maximally retains information about the input sentence $x$, while minimizing the mutual information with protected attributes $a$. This objective aligns with the information bottleneck principle, and is formally expressed as equation (7), where $I$ represents mutual information, $X$ is the original embedded representation, $R$ is the mutated latent vector, $A$ is the attribute of protected feature, and $\beta$ is a coefficient to balance the two terms.

$$\max \mathcal{L} = I\left(R; X\right) - \beta I\left(R; A\right) \tag{7}$$

To deal with the difficulty in estimating mutual information, this paper uses upper and lower bounds along with Monte Carlo gradient estimates. This paper introduces the following lemmas, and the proofs for them are presented in Appendix G.

As maximizing $I(R; X)$ represents the association value between the original token $X$ and the latent vector $R$, we can maximize the lower bound of equation (8).

**Lemma 1.** *For any conditional distribution p,*

$$I\left(R; X\right) \geq \mathbb{E}_{x''_{(l,i)} \sim P\left(x''_{(l,i)}\right), r \sim m(r|z), z \sim e_1(z|x_{(l,i)})} \left[log D_2(x''_{(l,i)}|r)\right] + H(x) \tag{8}$$

*where $D_2$ is the function of decoder 2 $d_2$, q is the encoder function, p is the real distribution, m is the projection function, and H is the entropy.*

As entropy is non-negative, the lemma implies the lower bound as equation (9).

$$I\left(R; X\right) \geq \mathbb{E}_{x''_{(l,i)} \sim P\left(x''_{(l,i)}\right), r \sim m(r|z), z \sim e_1(z|x_{(l,i)})} \left[log D_2(x''_{(l,i)}|r)\right] := L_r \tag{9}$$

At the same time, by minimizing $I(R; A)$ which represents the association value between the latent vector $R$ and the protected feature attribute $A$, we can also maximize the lower bound of equation (7).

**Lemma 2.** *For any conditional distribution m,*

$$I\left(R; A\right) \leq \int m\left(r|z\right) p\left(z\right) log \frac{m(r|z)}{s(r)} := C_1 \tag{10}$$

However, it is not ideal because considering only the function $m$ is not stable. For the tighter upper bound when assuming optimal dual regularization, $I(R; A)$ can be approximated.

**Corollary 3.** *If $\mathcal{D}_{KL}(p(a|r) \parallel h(a|r)) \leq l$,*

$$I\left(R; A\right) \leq \mathbb{E}_{p(r,a)} \left[\log p\left(a \mid r\right) - \log p\left(a\right)\right] + l \tag{11}$$

As the adversarial discriminator approaches the global optimum, that is, $l \to 0$, the upper bound of $I(R; A)$ can be written as equation (12).

$$I\left(R; A\right) \leq \mathbb{E}_{p(r,a)} \left[\log p\left(a \mid r\right) - \log p\left(a\right)\right] := C_2 \tag{12}$$

The final objective function for bias mitigation can be formulated as equation (7).

$$\max \mathcal{L}_r - \beta_1 C_1 - \beta_2 C_2 \tag{13}$$

As the autoencoder is optimized in the distribution space, this paper can show that strong duality holds within a constrained optimization approach (Song et al., 2019),

**Theorem 4.** *If $\epsilon_1, \epsilon_2, \epsilon_3 > 0$, then strong duality holds for the following optimization problem over distributions p,$e_1$,m:*

$$\min_{p,e_1,m} L \quad s.t. C_1 < \epsilon_1, C_2 < \epsilon_2 \tag{14}$$

---

**Algorithm 1:** Latent Regularization via Adversarial Training

**Data:** Data $X$ and protected features $A$
**Result:** Fully trained encoder $e_1$
Initialize $e_1, D_1, D_2$;
**for** *epochs* **do**
  **for** *batches* **do**
    Sample $x, a, \widetilde{a}$ from $X, A$, $U(0,1)$ respectively;
    $\theta_q \leftarrow \theta_q - \eta \frac{\partial \mathcal{L}_r}{\partial \theta_{e_1}}(x, \widetilde{a})$;
    $\theta_{D_1} \leftarrow \theta_{D_1} - \eta \frac{\partial \mathcal{L}_r}{\partial \theta_{D_1}}(x, \widetilde{a})$;
    $\theta_{e_1} \leftarrow \theta_{e_1} - \eta \frac{\partial (C_1 + C_2)}{\partial \theta_{e_1}}(x, \widetilde{a})$;
    $\theta_m \leftarrow \theta_m - \eta \frac{\partial (C_1 + C_2)}{\partial \theta_m}(x, \widetilde{a})$;
    $\theta_{D_2} \leftarrow \theta_{D_2} - \eta \frac{\partial C_2}{\partial \theta_{D_2}}(x, a)$;
  **end**
**end**
**return** $e_1$;

---

While $\mathcal{L}_r$ encourages $z$ and $r$ to retain maximal information about the input $x$, the regularization terms $C_1$ and $C_2$ act as adversarial constraints, discouraging encoding of protected attribute information. Algorithm 1 shows its training process.

### 3.3 USING DEBIASED LLMS FOR DOWNSTREAM TASKS

MQAR functions as a lightweight plug-in module applied at the self-attention level of a pre-trained LLM, transforming token representations before they are processed by the frozen layers. The transformation preserves semantic content while suppressing protected attribute signals. Once debiased, the modified embeddings are propagated unchanged through the rest of the model. Since MQAR does not require access to backbone weights or gradient updates, it is well-suited for deployment in black-box models and resource-constrained environments.

To train the adversarial discriminator $D_a$, MQAR adopts weak supervision from curated lexicons (Dev et al., 2020), selecting only instances that contain clear linguistic cues of protected attributes (e.g., pronouns, occupational terms). Ambiguous or neutral instances are excluded to prevent supervision noise and maintain precise regularization. This selective supervision ensures that bias mitigation is applied only where necessary, avoiding distortion of unbiased content.

Unlike prior debiasing approaches that rely on model fine-tuning, data augmentation, or training-time fairness constraints, MQAR operates entirely during inference. It introduces no structural modifications and requires no retraining, making it a practical and scalable solution for fairness-sensitive applications.

## 4 EXPERIMENTAL RESULTS

### 4.1 EXPERIMENTAL SETUP

For experiments, this paper has conducted all experiments on Ubuntu servers equipped with Intel(R) Xeon(R) Silver 4210R CPUs and four NVIDIA A100 GPUs. For each backbone LLM (BERT, T5, GPT-Neo, Mixtral, and LLaMA 3.2), this paper follows the original implementation and hyperparameter settings reported in the corresponding papers unless otherwise noted. The MQAR autoencoder is configured so that its input and output dimensions match the hidden size of each LLM, and the latent dimension is set to 400. This paper optimizes MQAR with a learning rate of $1 \times 10^{-4}$, apply early stopping with a patience of 7 based on validation loss (checked every 30 epochs), and keep all LLM parameters frozen.

MQAR is trained with multi-hot protected-attribute labels (e.g., gender and race) in the adversarial discriminator to jointly debias multiple attributes within a single model. Unless stated otherwise, this paper reports the mean and standard deviation over three random seeds for all metrics, and this paper uses paired $t$-tests to assess the statistical significance of improvements over baselines. Detailed hyperparameters and additional implementation details are provided in Appendix B.

### 4.2 RESULTS ON BIAS IN LLMS

To measure bias in LLMs, this paper uses WinoBias (Zhao et al., 2018), StereoSet (Nadeem et al., 2020), CrowS-Pairs (Nangia et al., 2020). This section compares the raw LLMs against a wide range of debiasing methods, including representation-level approaches (INLP (Ravfogel et al., 2020), SentDebias (Liang et al., 2020)), model-level methods (ADELE (Lauscher et al., 2021), FaRM (Chowdhury & Chaturvedi, 2022), FineDeb(Saravanan et al., 2023)), recent deep debiasing modules (HDD (Zayed et al., 2023)), and inference-time debiasing techniques (CRISPR(Yang et al., 2024), RB(Kim et al., 2025)), in addition to the MQAR. Appendix H summarizes the datasets and metrics in detail.

Figure 3 reports WinoBias results across five LLMs. Pro (pro-stereotype) and Anti (anti-stereotype) denote the intensities of pro- and anti-stereotypical preferences, respectively. Avg. is their average, and $|Diff|$ is the absolute difference, which this experiment uses as a bias score. As shown, INLP, SentDebias, ADELE, FaRM, FineDeb, HDD, and MQAR all reduce stereotypical associations compared to the raw LLMs. Among them, MQAR achieves the lowest Avg. and a notably small $|Diff|$, indicating that it better balances stereotype reduction with context preservation.

Table 1 summarizes results on StereoSet and CrowS-Pairs for LLaMA 3.2 with different debiasing methods, results for other backbones are provided in Appendix C and results with standard deviation are provided in Appendix E. Representation-level methods such as INLP and SentDebias reduce the stereotype score (SS) and bias on CrowS-Pairs but often yield suboptimal ICAT or

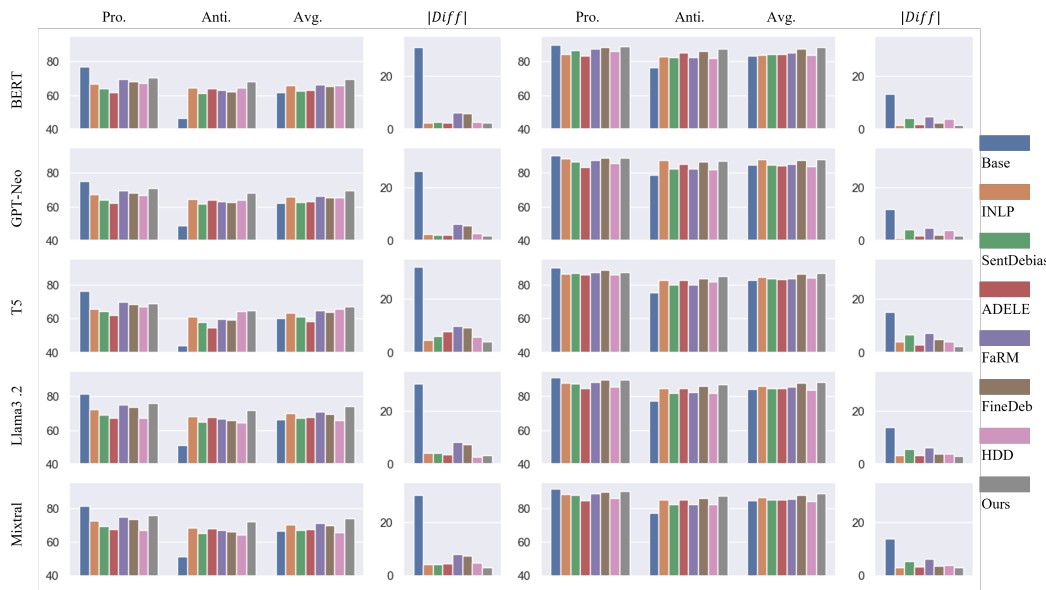

Figure 3: Experimental results of WinoBias. Pro and Anti show intensity of pro-stereotype and anti-stereotype, respectively. Avg and $|Diff|$ show average and difference of both, which show the bias score of LLMs.

Table 1: Experimental results of StereoSet (Nadeem et al., 2020), CrossPairs (Nangia et al., 2020). LMS (language modeling score), SS (stereotype score), and ICAT (idealized CAT score) are bias metrics of StereoSet. All (all dataset), Stereo (stereo dataset), and Anti (anti-stereo dataset) are bias metrics of CrowS-Pairs. First place is written in bold.

| Model | StereoSet | | | | | | CrowS-Pairs | | |
| | Gender | | | Race | | | Gender | | |
| | LMS | SS | ICAT | LMS | SS | ICAT | All | Stereo | Anti. |
|---|---|---|---|---|---|---|---|---|---|
| Llama 3.2 | 89.9 | 59.7 | 72.5 | 90.1 | 62.1 | 68.3 | 61.9 | 62.1 | 60.7 |
| +INLP | 90.6 | 58.3 | 75.6 | 90.4 | 64 | 65 | 48.2 | 48.6 | **45.9** |
| +SentDebias | 91.5 | 60.2 | 72.8 | 90.8 | 56.3 | 79.4 | 51.3 | 52.2 | 46 |
| +ADELE | 90.7 | 58.2 | 75.8 | 90.5 | 64.1 | 65.1 | 48.3 | 48.7 | 45.9 |
| +FaRM | 91.4 | 60.3 | 72.6 | 90.8 | 56.2 | 79.5 | 51.4 | 52.2 | 46.1 |
| +FineDeb | 82.1 | 54.2 | 75.2 | 77 | **51.8** | 74.1 | 57.1 | 57.1 | 57.3 |
| +CRISPR | 89.7 | 60.3 | 72.4 | 91.0 | 53.2 | 81.5 | 50.8 | 52.0 | 47.1 |
| +RB | 90.8 | 54.1 | 76.7 | 89.1 | 54.7 | 82.8 | 49.7 | 52.3 | 48.3 |
| +Ours | **91.6** | **53.1** | **86** | **91.3** | 54 | **84** | **48.1** | **47.9** | 49.9 |

All scores. Model-level debiasing (ADELE, FaRM, FineDeb) and recent deep debiasing modules (HDD) further improve certain metrics at the cost of increased computational overhead. Inference-time baselines such as CRISPR offer modest bias reductions without modifying model parameters, but they largely leave the underlying attention representations unchanged.

By contrast, MQAR achieves the best or near-best scores across LMS, SS, ICAT, and CrowS-Pairs (All/Stereo/Anti) on LLaMA 3.2. This indicates that quantized attention regularization effectively suppresses stereotypical associations while preserving language modeling quality, outperforming both representation-level and inference-time baselines on these benchmarks.

## 4.3 RESULTS ON DIFFERENT DOWNSTREAM TASKS

This paper next assesses whether MQAR preserves utility while mitigating bias on three downstream tasks: abusive language detection, hate speech detection, and sentiment analysis, as well as a text

Table 2: Experimental results for downstream tasks. AUC, FPED, and FNED are represented as percentage score. First place written in bold and second place written in underlined.

| Task | Type | Metric | Model | | | | |
| --- | --- | --- | --- | --- | --- | --- | --- |
| | | | Baseline | OSCaR | SentDebais | INLP | Ours |
| Abusive language detection (Founta) | Original Data | AUC | **93.8** | 93.5 | 93.6 | 93.7 | 93.7 |
| | | FPED | 2.32 | **1.20** | 2.53 | 1.92 | 1.87 |
| | | FNED | 3.71 | 6.21 | 3.46 | 6.34 | **3.44** |
| | Generated Data | AUC | 92.3 | 91.9 | 92.5 | 91.5 | **92.8** |
| | | FPED | 0.262 | 0.654 | 0.131 | 0.314 | **0.0654** |
| | | FNED | 0.251 | **0.036** | 0.0835 | 0.332 | 0.167 |
| Hate speech detection (CMSB) | Original Data | AUC | **96.5** | 94.3 | 96.3 | 88.2 | 95.1 |
| | | FPED | 0.121 | 0.443 | **0.0117** | 0.502 | 0.060 |
| | | FNED | 9.54 | 3.61 | 4.43 | 12.4 | **3.21** |
| | Generated Data | AUC | 94.7 | 89.2 | **94.9** | 84.8 | 94.3 |
| | | FPED | 0.0584 | 0.0562 | **0.0137** | 0.0192 | 0.0188 |
| | | FNED | 3.01 | 1.01 | 0.0442 | 0.0257 | **0.0218** |
| Sentiment analysis (EEC) | Anger Emotion | $\Delta_{F\uparrow - M\downarrow}$ | 0.0074 | 0.0092 | 0.0121 | **0.0052** | **0.0052** |
| | | $\Delta_{F\downarrow - M\uparrow}$ | 0.0316 | 0.0217 | **0.0149** | 0.0175 | 0.0163 |
| | Anger Valence | $\Delta_{F\uparrow - M\downarrow}$ | 0.0219 | 0.0159 | 0.0130 | **0.0121** | 0.0133 |
| | | $\Delta_{F\downarrow - M\uparrow}$ | 0.0198 | 0.0137 | 0.0119 | 0.0130 | **0.0105** |
| Text generation | GPTScore | Bias | 8.73 | 4.36 | 5.82 | 4.31 | **4.21** |
| | BLUE | Bias | 0.1 | **0.07** | 0.11 | 0.08 | 0.08 |
| Question answering | | Bias | 36.8 | 33.5 | 35.3 | 34.6 | 35.5 |

generation setting. For these tasks, this experiment focuses on gender bias and compare a GPT-Neo backbone equipped with OSCaR, SentDebias, INLP, and MQAR against the original GPT-Neo baseline. All results are averaged over three random seeds . Overall, MQAR achieves an average accuracy drop of at most $0.4$ percentage points relative to the non-debiased baseline across all tasks (see Table 2 and Appendix D).

Experiments of abusive language detection are conducted on Twitter datasets from Founta et al. (2018). The experimental results are evaluated on two types of datasets: original dataset and generated dataset. The former shows how bias appears against the original data distribution that is close to the real situation. The latter is constructed with the selection of sentences containing female and male phrases from the original datasets and the addition of those replacing female and male phrases, respectively. The generated dataset that gives the distributions of sentences containing female and male phrases has the same bias distribution. FPED and FNED are used as bias metrics, and AUC is used as accuracy (Park et al., 2018). FPED and FNED are calculated as equation (15) where FPR is a false positive rate, FNR is a false negative rate and T is a set of all groups. The upper side of Table 2 shows the results.

$$FPED = \Sigma_{t \in T}|FPR - FPR_t|, FNED = \Sigma_{t \in T}|FNR - FNR_t| \tag{15}$$

OSCaR achieves strong reductions in FPED and FNED but at the cost of a noticeable decrease in AUC, indicating that aggressive projection can remove task-relevant information. SentDebias maintains higher AUC but suffers from information loss due to its projection-based debiasing. By contrast, MQAR attains AUC that is statistically indistinguishable from the baseline while achieving competitive or better FPED/FNED scores, showing that it can reduce bias with only a small accuracy drop.

Sentiment analysis using EEC is evaluated based on the difference in emotion prediction values between sentences with male phrases and female phrases, where low numbers mean low bias. The anger attribute is identified mainly where gender bias is the most serious. The middle of Table 2 shows its result. The proposed model has lower numerical values than other methods, illustrating that the model effectively removes the bias even for the more difficult downstream tasks.

In the case of text generation, this study utilizes the bias measurement method proposed by Sun et al. This method involves scaling the metrics used to evaluate text generation, and then using the differences in each attribute to measure bias levels. Although this method can be applied to various metrics, in this paper, it is implemented using BLEU and GPTScore. The experimental results are summarized in the lower part of Table 2. Compared with other methods, it is confirmed that the approach presented in this paper is effective in appropriately mitigating bias.

## 4.4 EFFICIENCY AND OVERHEAD

This experiment analyzes the computational overhead introduced by MQAR. Table 3 reports the additional parameters, FLOPs per forward pass, and end-to-end inference latency for each backbone LLM with different debiasing methods. On LLaMA 3.2 with StereoSet, MQAR adds only 48.1% parameters relative to the backbone and increases FLOPs by 87.0%, yet it achieves approximately 43% lower end-to-end inference time than FineDeb, which requires full-model fine-tuning. Similar trends hold for other backbones, where MQAR consistently offers a favorable trade-off between fairness improvements and computational cost due to its frozen-backbone design and lightweight quantization modules.

Table 3: Computational overhead and latency on Llama 3.2. Param and FLOPs denote relative overhead compared to the frozen backbone (in %), and Latency reports end-to-end inference time per 100 generated tokens (in ms).

| Method | Param | FLOPs | Latency |
|---|---|---|---|
| Backbone | 0.0 | 0.0 | 97.8 |
| FineDeb | +107.3 | +191.8 | 189.8 |
| MQAR | +48.1 | +87.0 | 132.7 |

## 4.5 ABLATION STUDY

For the proposed method, this paper checks the results of the absence of the discriminator and decoder to confirm the effect of each component on bias mitigation. An experiment is conducted by checking the metric values for abusive language detection with the BERT model. Table 4 illustrates the result.

If the decoder is removed, the original data cannot be maintained during bias mitigation, resulting in a significant decrease in accuracy and an increase in the bias metric due to data loss. As a result, it can be confirmed that the decoder plays an important role in maintaining the original information. If the discriminator is removed, the original data are retained, but bias mitigation is not done properly, resulting in high bias for the dataset.

Table 4: Ablation study on abusive language detection. (a) is the result of decoder removal and (b) is that of discriminator.

| Dataset | Metric | Original | (a) | (b) |
|---|---|---|---|---|
| Original | AUC | 93.9 | 50.3 | 93.9 |
| | FPED | 1.84 | 20.2 | 2.50 |
| | FNED | 3.46 | 18.3 | 3.46 |
| Generated | AUC | 92.8 | 48.9 | 92.4 |
| | FPED | 0.0654 | 22.1 | 0.392 |
| | FNED | 0.167 | 17.5 | 0.250 |

## 5 CONCLUSION

This paper proposes MQAR, a model-agnostic debiasing method that mitigates multi-attribute social bias in frozen LLMs via structured quantized regularization. Unlike existing approaches that require fine-tuning or data augmentation, MQAR directly operates on self-attention layers, disentangling protected attribute activations while preserving semantic content. Extensive experiments across five LLMs and three standard bias benchmarks show that MQAR achieves consistent bias reduction while maintaining downstream task accuracy within 0.4% of original models.

While MQAR demonstrates strong performance in zero-shot and large-data settings, future work can explore its application in low-resource regimes where robust disentanglement is more challenging. Moreover, extending MQAR to multilingual models and instruction-tuned LLMs presents promising directions for broadening fairness-aware NLP. Overall, MQAR provides a lightweight, scalable solution for fair language modeling, pushing the frontier toward bias-resilient LLM deployment.

ACKNOWLEDGMENTS

This work was supported by the Yonsei Fellow Program funded by Lee Youn Jae, IITP grant funded by the Korea government (MSIT) (RS-2022-II220113, Developing a Sustainable Collaborative Multi-modal Lifelong Learning Framework), and Air Force Defense Research Sciences Program funded by Air Force Office of Scientific Research.

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

Table 5: Parameters for Training BERT, GPT, Llama, T5 and Mixtral

| Model | Parameters | Value |
|---|---|---|
| BERT | Model name | bert-base-uncased |
| | EPS | 1.00E-15 |
| | latent vector dimension | 400 |
| | learning rate for encoder/decoder | 1.00E-04 |
| | learning rate for discriminator/classifier | 5.00E-05 |
| | learning rate for fine-tuning BERT on downstream task | 2.00E-05 |
| | random seed | 0 |
| | patience for early stop | 7 |
| GPT | Model name | |
| | EPS | 1.00E-05 |
| | latent vector dimension | 1000 |
| | learning rate for encoder/decoder | 5.00E-04 |
| | learning rate for discriminator/classifier | 5.00E-05 |
| | learning rate for fine-tuning GPT on downstream task | 1.00E-05 |
| | random seed | 0 |
| | patience for early stop | 7 |
| Llama | Model name | Llama-3.2-3b |
| | EPS | 1.00E-15 |
| | latent vector dimension | 800 |
| | learning rate for encoder/decoder | 1.00E-04 |
| | learning rate for discriminator/classifier | 5.00E-05 |
| | learning rate for fine-tuning Llama on downstream task | 2.00E-05 |
| | random seed | 0 |
| | patience for early stop | 7 |
| T5 | Model name | t5-base |
| | EPS | 1.00E-06 |
| | latent vector dimension | 400 |
| | learning rate for encoder/decoder | 1.00E-04 |
| | learning rate for discriminator/classifier | 5.00E-05 |
| | learning rate for fine-tuning T5 on downstream task | 1.00E-04 |
| | random seed | 0 |
| | patience for early stop | 7 |
| Mixtral | Model name | Mixtral-8x7B-v0.1 |
| | EPS | 1.00E-06 |
| | latent vector dimension | 400 |
| | learning rate for encoder/decoder | 1.00E-04 |
| | learning rate for discriminator/classifier | 5.00E-05 |
| | learning rate for fine-tuning Mixtral on downstream task | 1.00E-04 |
| | random seed | 0 |
| | patience for early stop | 7 |

## A  PROTECTED-ATTRIBUTE LEXICONS AND SUPERVISION

This appendix describes how the proposed method constructs and validates the weak supervision signals, focusing on lexicons for gender and race, and classifier-based labels for other protected attributes such as religion and region.

**Gender and race lexicons.**  For gender and race, this paper follows prior work on fairness in language models and sentiment/toxicity/abusive classification, which commonly uses demographic term lists as weak supervision signals for protected attributes (e.g., gendered pronouns, kinship terms, and occupational titles).  Building on these sources, this paper constructs moderate-sized lexicons consisting of gendered and race-related terms that frequently appear in the corpora.

For gender, this paper uses a list of masculine and feminine terms covering pronouns, kinship relations, professions, and other common descriptors. The full list used in the experiments is shown below:

- **Male:** he, him, masculine, actor, author, boy, bridegroom, brother, conductor, count, czar, daddy, duke, man, emperor, father, grandfather, heir, host, husband, king, master, murderer, nephew, poet, policeman, prince, sir, son, uncle, wizard, waiter, guy, boyfriend, dad, gentleman, lord, monk, priest, prophet, patron, viscount, shepherd, steward, baron, peer, abbot, traitor, benefactor, hunter, tempter, enchanter, songster, manservant, landlord, milkman, giant.

- **Female:** she, her, feminine, actress, authoress, girl, bride, sister, conductress, countess, czarina, mummy, duchess, woman, empress, mother, grandmother, heiress, hostess, wife, queen, mistress, murderess, niece, poetess, policewoman, princess, madam, daughter, aunt, witch, waitress, girlfriend, mom, lady, nun, priestess, prophetess, patroness, viscountess, shepherdess, stewardess, baroness, peeress, abbess, traitress, benefactress, huntress, temptress, enchantress, songstress, maidservant, landlady, milkmaid, giantess.

For race and ethnicity, this paper constructs a lexicon of descriptors (e.g., ethnic and regional identifiers, demonyms) by aggregating terms from prior demographic-bias benchmarks and public demographic resources, followed by manual filtering to remove rare or ambiguous entries. Due to space constraints, this paper does not list all race-related terms here; representative examples and full vocabularies will be released with the code.

**Filtering overlaps with target labels.** To avoid leakage between protected-attribute supervision and downstream task labels, this paper explicitly removes lexicon entries that coincide with target label tokens. Concretely, for each downstream dataset (e.g., abusive language and hate speech detection), this paper (i) constructs the set of label strings and their tokenization, (ii) automatically drops any lexicon term that appears verbatim in the label vocabulary, and (iii) manually inspects borderline cases where a lexicon term could be semantically close to a label (e.g., "offensive", "toxic") and removes them if necessary. This procedure ensures that the proposed protected-attribute supervision does not directly encode task-specific labels.

**Classifier-based supervision for other attributes.** For attributes that are less amenable to stable lexicon construction—such as religion, region, or other socio-demographic categories—this paper resorts to off-the-shelf sentence-level classifiers. Specifically, this paper applies publicly available pre-trained models (e.g., toxicity/fairness or demographic-attribute classifiers) to each sentence and treats high-confidence predictions as weak labels for the corresponding protected attribute. This paper then binarizes these predictions into multi-hot attribute vectors and uses them in the adversarial discriminator. In all cases, this paper only retains predictions above a conservative confidence threshold to reduce noise and treats this classifier-based supervision as complementary to lexicon-based labels rather than a replacement.

Overall, this hybrid strategy—combining curated lexicons for gender and race with classifier-based supervision for other attributes, and explicitly filtering out overlaps with downstream labels—provides a flexible yet controlled way to obtain protected-attribute signals needed to train MQAR without requiring manual annotation of sensitive attributes.

## B  HYPERPARAMETERS FOR TRAINING LLMS

The main parameters of the model are summarized in Table 1 and 2. For most hyperparameters, the default values are set based on the papers proposing each LLM. The autoencoder of the proposed model is designed as a Multi-Layer Perceptron (MLP) using three layers. Experiments are conducted by adjusting the dimension of each latent vector and the learning rate. Training is incorporated with early stopping, and the related numerical values are also documented in the table. For the downstream tasks, the model has been fine-tuned using each dataset.

Table 6: Experimental results of StereoSet (Nadeem et al., 2020), CrossPairs (Nangia et al., 2020). LMS (language modeling score), SS (stereotype score), and ICAT (idealized CAT score) are bias metric of StereoSet. All (all dataset), Stereo (stereo dataset), and Anti (anti-stereo dataset) are bias metrics of CrowS-Pairs. First place is written in bold.

| Model | StereoSet | | | | | | CrowS-Pairs | | |
| | Gender | | | Race | | | Gender | | |
| | LMS | SS | ICAT | LMS | SS | ICAT | All | Stereo | Anti. |
|---|---|---|---|---|---|---|---|---|---|
| BERT-Base | 85.4 | 58.3 | 71.2 | 88.3 | 61.7 | 67.6 | 60.5 | 61.1 | 56.9 |
| BERT-Base+INLP | 86.3 | 57.3 | 73.7 | 88.9 | 63 | 65.8 | 48.3 | 49.7 | **40** |
| BERT-Base+SentDebias | 87.2 | 59.4 | 70.8 | 89.3 | 55.2 | 80 | 51.3 | 53.2 | 40.1 |
| BERT-Base+ADELE | 86.3 | 57.3 | 73.7 | 88.9 | 63 | 65.8 | 48.3 | 49.7 | 40 |
| BERT-Base+FaRM | 87.2 | 59.4 | 70.8 | 89.3 | 55.2 | 80 | 51.3 | 53.2 | 40.1 |
| BERT-Base+FineDeb | 77.7 | 53.3 | 72.6 | 75.4 | **50.8** | 74.1 | 54.6 | 58.1 | 51.5 |
| BERT-Base+Ours | **87.3** | **52.3** | **83.3** | **89.8** | 53.1 | **85.7** | 48.2 | 48.9 | 44.1 |
| T5 | 84.7 | 60.2 | 67.4 | 88 | 62.3 | 66.4 | 64.2 | 66.3 | 51.3 |
| T5+INLP | 85.3 | 58.6 | 70.6 | 88.4 | 64.3 | 63.1 | 50.5 | 52.8 | 36.4 |
| T5+SentDebias | 86.2 | 60.7 | 67.7 | 88.8 | 56.5 | 77.3 | 53.6 | 56.4 | 36.6 |
| T5+ADELE | 85.4 | 58.8 | 70.4 | 88.4 | 64.1 | 63.5 | **50.4** | 52.7 | **36.3** |
| T5+FaRM | **86.4** | 60.8 | 67.8 | 88.8 | 56.3 | 77.6 | 53.4 | 56.3 | 36.5 |
| T5+FineDeb | 76.9 | 54.7 | 69.7 | 74.8 | **52.1** | 71.7 | 59.3 | 61.3 | 47.8 |
| T5+Ours | 86.4 | **53.8** | **79.9** | **89.2** | 54.3 | **81.5** | 50.4 | **52** | 40.6 |
| GPT-Neo | 86.1 | 59.5 | 70.2 | 89.1 | 60.8 | 71.3 | 63.0 | 65.5 | 53.1 |
| GPT-Neo+INLP | 86.6 | 57.5 | 74.1 | 89.2 | 63.3 | 66.0 | 48.7 | 50.1 | 40.3 |
| GPT-Neo+SentDebias | 87.4 | 59.6 | 71.1 | 89.6 | 55.6 | 80.4 | 51.5 | 53.5 | 40.4 |
| GPT-Neo+ADELE | 86.6 | 57.7 | 74.0 | 89.2 | 63.3 | 66.1 | 48.6 | 49.9 | **40.2** |
| GPT-Neo+FaRM | 87.5 | 59.7 | 71.1 | 89.7 | 55.4 | 80.4 | 51.6 | 53.5 | 40.5 |
| GPT-Neo+FineDeb | 78.0 | 53.6 | 73.0 | 75.6 | **51.1** | 74.4 | 54.9 | 58.4 | 51.7 |
| GPT-Neo+Ours | **87.7** | **52.5** | **83.6** | **90.1** | 53.4 | **86.0** | **48.4** | **49.2** | 44.4 |
| Mixtral | 86.9 | 58.5 | 72.2 | 89.9 | 59.8 | 73.3 | 61.8 | 64.3 | 52.5 |
| Mixtral+INLP | 87.4 | 56.5 | 76.1 | 90.0 | 62.3 | 68.0 | 47.5 | 48.9 | 39.7 |
| Mixtral+SentDebias | 88.2 | 58.6 | 73.1 | 90.4 | 54.6 | 82.4 | 50.3 | 52.3 | 39.8 |
| Mixtral+ADELE | 87.4 | 56.7 | 76.0 | 90.0 | 62.3 | 68.1 | 47.4 | 48.7 | **39.6** |
| Mixtral+FaRM | 88.3 | 58.7 | 73.1 | 90.5 | 54.4 | 82.4 | 50.4 | 52.3 | 39.9 |
| Mixtral+FineDeb | 78.8 | 52.6 | 75.0 | 76.4 | **50.1** | 76.4 | 53.7 | 57.2 | 51.1 |
| Mixtral+Ours | **88.5** | **51.5** | **85.6** | **90.9** | 52.4 | **88.0** | **47.2** | **48.0** | 43.8 |

## C    RESULT ON BIAS IN OTHER LLMS

Table 3 shows the result of two datasets, StereoSet (Nadeem et al., 2020) and CrossPairs (Nangia et al., 2020), with four LLMs, BERT, T5, GPT-Neo, and Llama 3.2. From these metrics, the experimental results show that, similar as the content of the main paper, INLP, SentDebias, ADELE, FaRM, and FineDeb have mitigated stereotypical associations compared to the original models. The proposed method demonstrated superior performance in these two datasets compared to other algorithms, suggesting that it is more effective in preserving semantic elements and removing biased elements in stereotype categories than other algorithms.

## D    RESULT ON DOWNSTREAM TASKS IN OTHER LLMS

Table 4 shows the result of 4 downstream tasks, abusive language detection (Founta et al., 2018), sentiment analysis (Kiritchenko & Mohammad, 2018), text generation (Sun et al., 2019), question answering (Hendrycks et al., 2020), with two LLMs, BERT and T5 (the result of GPT-Neo is written in section 4.3).

Table 7: Experimental results for downstream tasks. AUC, FPED, and FNED are represented as percentage score. First place written in bold and second place written in underlined.

| Task | Type | Metric | Model | | | | |
|---|---|---|---|---|---|---|---|
| | | | Baseline | OSCaR | SentDebais | INLP | Ours |
| BERT | | | | | | | |
| Abusive language detection (Founta) | Original Data | AUC | **93.8** | 93.5 | 93.6 | 93.7 | 93.7 |
| | | FPED | 2.32 | **1.20** | 2.53 | 1.92 | 1.87 |
| | | FNED | 3.71 | 6.21 | 3.46 | 6.34 | **3.44** |
| | Generated Data | AUC | 92.3 | 91.9 | 92.5 | 91.5 | **92.8** |
| | | FPED | 0.262 | 0.654 | 0.131 | 0.314 | **0.0654** |
| | | FNED | 0.251 | **0.036** | 0.0835 | 0.332 | 0.167 |
| Hate speech detection (CMSB) | Original Data | AUC | **96.5** | 94.3 | 96.3 | 88.2 | 95.1 |
| | | FPED | 0.121 | 0.443 | **0.0117** | 0.502 | 0.060 |
| | | FNED | 9.54 | 3.61 | 4.43 | 12.4 | **3.21** |
| | Generated Data | AUC | 94.7 | 89.2 | **94.9** | 84.8 | 94.3 |
| | | FPED | 0.0584 | 0.0562 | **0.0137** | 0.0192 | 0.0188 |
| | | FNED | 3.01 | 1.01 | 0.0442 | 0.0257 | **0.0218** |
| Sentiment analysis (EEC) | Anger Emotion | $\Delta_{F\uparrow-M\downarrow}$ | 0.0074 | 0.0092 | 0.0121 | **0.0052** | **0.0052** |
| | | $\Delta_{F\downarrow-M\uparrow}$ | 0.0316 | 0.0217 | **0.0149** | 0.0175 | 0.0163 |
| | Anger Valence | $\Delta_{F\uparrow-M\downarrow}$ | 0.0219 | 0.0159 | 0.0130 | **0.0121** | 0.0133 |
| | | $\Delta_{F\downarrow-M\uparrow}$ | 0.0198 | 0.0137 | 0.0119 | 0.0130 | **0.0105** |
| Text generation | BERTScore | Bias | 8.73 | 4.36 | 5.82 | 4.31 | **4.21** |
| | BLUE | Bias | 0.1 | **0.07** | 0.11 | 0.08 | 0.08 |
| T5 | | | | | | | |
| Abusive language detection (Founta) | Original data | AUC | **91.4** | 90.1 | 89.7 | 90.7 | 90.9 |
| | | FPED | 2.45 | **1.13** | 2.61 | 2.03 | 1.72 |
| | | FNED | 4.03 | 5.7 | 3.19 | 6.66 | **3.17** |
| | Generated data | AUC | 89.4 | **90.7** | 89.3 | 90.1 | 90.5 |
| | | FPED | 0.256 | 0.605 | 0.123 | 0.335 | **0.071** |
| | | FNED | 0.243 | **0.038** | 0.187 | 0.32 | 0.183 |
| Sentiment analysis (EEC) | Anger emotion | $\Delta_{F\uparrow-M\downarrow}$ | 0.0076 | 0.0088 | 0.0132 | **0.0049** | 0.0053 |
| | | $\Delta_{F\downarrow-M\uparrow}$ | 0.0287 | 0.0214 | 0.0139 | 0.0165 | **0.0117** |
| | Anger valence | $\Delta_{F\uparrow-M\downarrow}$ | 0.0199 | 0.0144 | 0.0132 | 0.013 | 0.0121 |
| | | $\Delta_{F\downarrow-M\uparrow}$ | 0.0187 | 0.0134 | 0.0121 | 0.0123 | **0.0107** |
| Text generation | GPTScore | Bias | 5.51 | 4.12 | 4.48 | 4.36 | **4.01** |
| | BLUE | Bias | 0.09 | 0.07 | 0.1 | 0.09 | **0.08** |
| Question answering | | Bias | **29.1** | 26.9 | 24 | 24.3 | 28.8 |

## E    RESULTS ON BIAS IN LLaMA WITH STANDARD DEVIATION

In the main paper (Table 1), this paper reported bias results on StereoSet and CrowS-Pairs for LLaMA 3.2 and several debiasing methods using single-point estimates. For completeness, this section provides the corresponding results with mean ± standard deviation over multiple random seeds, restricted to the LLaMA 3.2 backbone. This paper splits the results by dataset for readability: Table 8 reports StereoSet scores, and Table 9 reports CrowS-Pairs scores.

Across both benchmarks, this paper observes that the variance across runs is relatively small for all methods, indicating that the reported improvements are stable rather than artifacts of a single seed. On StereoSet, MQAR achieves the best or near-best ICAT scores while substantially reducing the stereotype score (SS) for both gender and race, without degrading the language modeling score (LMS). On CrowS-Pairs, MQAR attains the lowest (or competitive) bias scores on the All and Stereo subsets while keeping performance on the Anti subset comparable to other methods. These patterns are consistent with the main-table findings and support the claim that MQAR provides robust fairness gains on LLaMA 3.2 with minimal variance across runs.

Table 8: StereoSet results for LLaMA 3.2 with debiasing methods, reported as mean $\pm$ standard deviation over multiple runs. LMS, SS, and ICAT are language modeling, stereotype, and idealized CAT scores, respectively. Best values are in bold.

| Model | Gender | | | Race | | |
|---|---|---|---|---|---|---|
| | LMS | SS | ICAT | LMS | SS | ICAT |
| LLaMA 3.2 | 89.9±0.2 | 59.7±0.4 | 72.5±0.5 | 90.1±0.2 | 62.1±0.4 | 68.3±0.5 |
| +INLP | 90.6±0.2 | 58.3±0.3 | 75.6±0.4 | 90.4±0.2 | 64.0±0.3 | 65.0±0.4 |
| +SentDebias | 91.5±0.3 | 60.2±0.6 | 72.8±0.7 | 90.8±0.3 | 56.3±0.6 | 79.4±0.7 |
| +ADELE | 90.7±0.3 | 58.2±0.5 | 75.8±0.6 | 90.5±0.3 | 64.1±0.5 | 65.1±0.6 |
| +FaRM | 91.4±0.3 | 60.3±0.6 | 72.6±0.6 | 90.8±0.3 | 56.2±0.6 | 79.5±0.6 |
| +FineDeb | 82.1±0.6 | **54.2**±0.8 | 75.2±0.9 | 77.0±0.7 | **51.8**±0.8 | 74.1±0.9 |
| +CRISPR | 89.7±0.4 | 60.3±0.6 | 72.4±0.7 | 91.0±0.4 | 53.2±0.6 | 81.5±0.7 |
| +RB | 90.8±0.2 | 54.1±0.3 | 76.7±0.4 | 89.1±0.2 | 54.7±0.3 | 82.8±0.4 |
| +MQAR | **91.6**±0.2 | 53.1±0.3 | **86.0**±0.4 | **91.3**±0.2 | 54.0±0.3 | **84.0**±0.4 |

Table 9: CrowS-Pairs results for LLaMA 3.2 with debiasing methods, reported as mean $\pm$ standard deviation over multiple runs. All, Stereo, and Anti denote the overall, stereotype, and anti-stereotype subsets, respectively. Best values are in bold.

| Model | All | Stereo | Anti |
|---|---|---|---|
| LLaMA 3.2 | 61.9±0.6 | 62.1±0.6 | 60.7±0.6 |
| +INLP | 48.2±0.5 | 48.6±0.5 | **45.9**±0.5 |
| +SentDebias | 51.3±0.8 | 52.2±0.9 | 46.0±0.9 |
| +ADELE | 48.3±0.6 | 48.7±0.6 | 45.9±0.6 |
| +FaRM | 51.4±0.7 | 52.2±0.7 | 46.1±0.7 |
| +FineDeb | 57.1±1.1 | 57.1±1.2 | 57.3±1.0 |
| +CRISPR | 50.8±0.9 | 52.0±0.9 | 47.1±0.9 |
| +RB | 49.7±0.5 | 52.3±0.5 | 48.3±0.5 |
| +MQAR | **48.1**±0.4 | **47.9**±0.4 | 49.9±0.6 |

## F EFFECTIVENESS OF STRUCTURED QUANTIZATION

To isolate the effect of quantization in the proposed MQAR method, this section conducts an ablation study comparing three variants:

- **Full MQAR**: with both vector quantization and adversarial regularization.

- **No-Quant MQAR**: with adversarial regularization but without quantization.

- **Quant-only**: applying quantization without adversarial regularization.

This paper evaluates these variants on the StereoSet dataset using LLaMA 3.2. As shown in Table 10, Full MQAR achieves the lowest bias scores (SS) and the highest ICAT while slightly improving LMS over the baseline. Removing quantization (*No-Quant MQAR*) weakens attribute disentanglement and yields substantially worse ICAT, especially in scenarios with multiple protected attributes. Conversely, using quantization alone (*Quant-only*) reduces bias only marginally and even harms ICAT, indicating that the quantized bottleneck requires the adversarial regularizer to properly separate protected-attribute information from task-relevant semantics.

This section also studies the impact of the quantization codebook size $K$ in the latent space. In practice, this paper selects $K$ from a small candidate set ($K \in \{2, 3, 4, 8\}$) based on validation performance and keep $K$ fixed across layers for simplicity. The sensitivity analysis shows that MQAR is robust to moderate changes in $K$: very small codebooks (e.g., $K = 2$) overly compress the latent space and slightly hurt LMS, whereas very large codebooks weaken the debiasing bottleneck and increase SS. A moderate choice such as $K = 3$ consistently yields the best fairness-utility trade-off across benchmarks, and is therefore used in all main experiments.

Table 10: Effect of quantization on bias mitigation for LLaMA 3.2 on StereoSet (gender). LMS, SS, and ICAT denote language modeling, stereotype, and idealized CAT scores, respectively.

| Method | LMS | SS | ICAT |
|---|---|---|---|
| Full MQAR (ours) | **91.6** | **53.1** | **86.0** |
| No-Quant MQAR | 90.7 | 55.1 | 67.8 |
| Quant-only | 90.5 | 58.3 | 63.9 |
| Baseline (no mitigation) | 89.9 | 59.7 | 72.5 |
| $K = 2$ | 90.9 | 52.5 | 82.1 |
| $K = 3$ | **91.6** | **53.1** | **86.0** |
| $K = 4$ | 91.5 | 54.0 | 84.7 |
| $K = 8$ | 91.4 | 55.6 | 80.3 |

## G  OMITTED PROOF FOR LEMMAS AND THEOREM

**Lemma 5.** *For any conditional distribution $p$,*

$$I\left(R; X\right) \geq \mathbb{E}_{x^{''}_{(l,i)} \sim P\left(x^{''}_{(l,i)}\right), r \sim m(r|z), z \sim q(z|x_{(l,i)})}\left[log D_2(x^{''}_{(l,i)}|r)\right] + H(x_{(l,i)}) \qquad (16)$$

*where $D_2$ is the function of decoder 2 $d_2$, $q$ is the encoder function, $p$ is the real distribution, $m$ is the projection function, and $H$ is the entropy.*

*Proof.* As information is recovered with Decoder 2:

$$I\left(R; X\right) = \mathbb{E}_{z \sim q(z|x_{(l,i)}), r \sim m(r|z)}\left[\mathbb{E}_{x^{''}_{(l,i)} \sim p(x_{(l,i)}|r)}\left[log p(x^{''}_{(l,i)}|r)\right]\right] + H(x_{(l,i)}) \qquad (17)$$

$$\geq \mathbb{E}_{z \sim q(z|x_{(l,i)}), r \ m(r|z)}\left[\mathbb{E}_{x^{''}_{(l,i)} \sim p(x_{(l,i)}|r)}\left[log D_2(x^{''}_{(l,i)}|r)\right]\right] + H(x_{(l,i)}) \qquad (18)$$

$$= \mathbb{E}_{x^{''}_{(l,i)} \sim P\left(x^{''}_{(l,i)}\right), r \sim m(r|z), z \sim q(z|x_{(l,i)})}\left[log D_2(x^{''}_{(l,i)}|r)\right] + H(x_{(l,i)}) \qquad (19)$$

$\square$

**Lemma 6.** *For any conditional distribution $m$,*

$$I\left(R; A\right) \leq \int m\left(r|z\right) p\left(z\right) log \frac{m(r|z)}{s(r)} := C_1 \qquad (20)$$

*Proof.* Using additional function $m$, information is transformed from $z$ to $r$ and it is recovered with $D_2$.

$$I\left(R; A\right) = I\left(Z; D_2\left(m\left(R\right)\right)\right) \qquad (21)$$

$$\leq I\left(R; m\left(R\right)\right) \qquad (22)$$

$$\leq \int m\left(r|z\right) p\left(z\right) log \frac{m(r|z)}{s(r)} \qquad (23)$$

$\square$

**Corollary 7.** *If $\mathcal{D}_{KL}(p(a|r) \parallel h\left(a|r\right)) \leq l$,*

$$I\left(R; A\right) \leq \mathbb{E}_{p(r,a)}\left[\log p\left(a \mid r\right) - \log p\left(a\right)\right] + l \qquad (24)$$

*Proof.* For the tighter upper bound when assuming optimal dual regularization, $I(R; A)$ can be written as shown in Equations (25), where $\mathcal{D}_{KL}$ is KL divergence and $h$ is an any distribution.

$$I\left(R; A\right) = \mathbb{E}_{p(R,A)}\mathcal{D}_{KL}(p(a \mid r) \parallel h(a)) - \mathcal{D}_{KL}(p(a) \parallel h(a)) \qquad (25)$$

Where $\mathcal{D}_{KL} \geq 0$, it shows:

$$I\left(R; A\right) = \mathbb{E}_{p(R,A)} \mathcal{D}_{KL}(p(a \mid r) \parallel h(a)) - D_{KL}(p(a) \parallel h(a)) \tag{26}$$

Note that the distribution of $p$ is traceable which implies the Equations (27) and (29):

$$\mathbb{E}_{p(r,a)} \left[\log p\left(a \mid r\right) - \log p\left(a\right)\right] \tag{27}$$

$$= \mathbb{E}_{p(R,A)} \left[\mathcal{D}_{KL}(p(a \mid r) \parallel h(a)) - \mathcal{D}_{KL}(p(a|r) \parallel h\left(a|r\right))\right] \tag{28}$$

$$\geq \mathbb{E}_{p(R,A)} \mathcal{D}_{KL}(p(a \mid r) \parallel h(a)) - l \tag{29}$$

where $l$ is positive such that $\mathcal{D}_{KL}(p(a|r) \parallel h\left(a|r\right)) \leq l$. From Equation (8) and (11),

$$I\left(R; A\right) \leq \mathbb{E}_{p(R,A)} \left[\log p\left(a \mid r\right) - \log p\left(a\right)\right] + l \tag{30}$$

$\square$

## H  DETAILED INFORMATION ON DATASETS

The WinoBias dataset is the benchmark for paired male and female cross-reference-solving examples according to the Winograd format (Hirst, 1979; Peng et al., 2015). It contains two different subsets: the pro-stereotype where pronouns are anti-stereotypes primarily related to the gender-related occupation of pronouns, and the anti-stereotype when the opposite relationship is true. Each subset consists of two types of sentences. One is a sentence that requires a semantic understanding of the sentence for cross-reference resolution (Semantics Only), and the other is a sentence that relies on syntactic cues (w/ Semantic Cues). Gender bias is measured using the performance difference between a typical subset and a semi-structured subset.

StereoSet dataset is a benchmark dataset that measures the social bias of LLMs in that model. Gender and race are selected among the domains of it. LMS (language modeling score), SS (stereotype score), and ICAT (idealized CAT score) are bias metrics for StereoSet. The closer to 100 on LMS and ICAT, and the closer to 50 on SS, the more the bias is ideally mitigated. LMS is a value for the meaningful association, SS is a value for the stereotypic association, and ICAT represents a comprehensive number of these two figures. The CrowS-Pairs dataset is designed to evaluate social biases in textual models, containing pairs of sentences that highlight biases and stereotypes, in the CrowS-Pairs-stereo and CrowS-Pairs-antistereo categories. The dataset allows for the measurement of biases by comparing the model's responses to both biased and unbiased statements in the pairs.

## I  TABLE OF RELATED WORKS

Mitigating bias in large language models (LLMs) has received substantial attention, with prior works proposing methods across different stages of model deployment. Based on Table 11, this paper summarizes these approaches along three dimensions: model-level interventions, sentence-level representation manipulation, and prompt-based inference-time strategies.

**Model-level Approaches.** These methods directly modify the training process of LLMs. Projection-based techniques, such as SentDebias (Liang et al., 2020) and OSCaR (Dev et al., 2020), remove protected attribute components by projecting representations onto orthogonal subspaces. While simple, such methods risk reducing representation capacity for downstream tasks (Shin et al., 2020). Regularization-based methods, including FaRM (Chowdhury & Chaturvedi, 2022) and dropout-based debiasing (Webster et al., 2020), introduce fairness constraints into training losses. Others exploit token-level statistics, such as the masked token divergence in (Guo et al., 2022). However, all these methods typically require access to training data and full model fine-tuning, limiting scalability to new domains or closed-source LLMs.

Table 11: Related works for reducing bias of large language models (LLMs).

| Target | Method | Description |
|---|---|---|
| Learning Model | Projection based | Hard debiasing on layer result (Liang et al., 2020) |
| | | Projection on orthogonal subspace with correction and rectification (Dev et al., 2020) |
| | Regularization based | Drop out regularization (Webster et al., 2020) |
| | | Post-hoc debiasing procedure using intrinsic data (Schick et al., 2021) |
| | | Debiasing using the rate-distortion function (Chowdhury & Chaturvedi, 2022) |
| | [MASK] token based | Jensen–Shannon divergence with mask probability of pair sentences with protected feature (Guo et al., 2022) |
| | Data based | Generative self-conditioning methods (Cuadros et al., 2022) |
| | | Sustainable modular debiasing based on dedicated debiasing adapters with counterfactual data (Lauscher et al., 2021) |
| | | Combining important counterfactual (Zayed et al., 2023) |
| Embedded Sentence | Projection based | Interactive null space projection (Ravfogel et al., 2020) |
| | | Autoregressive interactive projection (Liang et al., 2021) |
| | Regularization based | Reinforced calibration on embedding (Liu et al., 2021) |
| | | LSTM-based bias mitigation on BERT (Pryzant et al., 2020) |
| Prompt of LLM | Static-prompt | Analyzing the bias mitigation effects of prompts in various levels of abstraction (Mattern et al., 2022) |
| | Dynamic-prompt | Using gender-neutral datasets for prompt to update biased word embeddings (Fatemi et al., 2021) |

**Embedding-level Techniques.**  These operate on the latent sentence representations of pre-trained models. INLP (Ravfogel et al., 2020) and its variants project sentence embeddings away from attribute-relevant directions. More recent work combines LSTM-based debiasing with BERT (Pryzant et al., 2020). However, such techniques may over-constrain the latent space, leading to information loss or degraded performance.

**Prompt-based Inference-time Strategies.**  These methods aim to mitigate bias without modifying model parameters, using static or dynamic prompts. Static prompt tuning (e.g., (Mattern et al., 2022)) adjusts prompt phrasing to reduce bias exposure. Dynamic prompt updates, such as those proposed by Fatemi et al. (2021), retrain embeddings with gender-neutral prompts or counterfactual data. While appealing due to their low-cost deployment, these methods cannot correct internal representational biases embedded within attention layers.

**Gap in Multi-Attribute Mitigation.**  Despite progress, most existing methods target single-attribute settings (e.g., gender only) and fail to generalize across intersecting attributes. Moreover, techniques that modify embeddings or prompts fail to address internal bias entanglement at the self-attention level—a known amplifier of representational bias (Jiang et al., 2022).

**Our Contribution.**  MQAR addresses these gaps by introducing a quantized regularization framework that operates directly on frozen self-attention layers. Unlike projection-based or prompt-based methods, MQAR disentangles and suppresses protected attribute signals within attention outputs, without requiring model fine-tuning. To our knowledge, it is the first scalable, multi-attribute debiasing method that is model-agnostic and operates entirely post-hoc, making it suitable for black-box LLM deployments.

