# OpenReview forum: "Multi-Feature Quantized Self-Attention for Fair Large Language Models"
_ICLR.cc/2026/Conference — ICLR 2026 Poster_

### Official Review · Reviewer_kQ3w · 2025-10-21

**Soundness:** 2
**Presentation:** 2
**Contribution:** 2
**Rating:** 4
**Confidence:** 3

**Summary:**

This paper proposes MQAR to solve the problem of social bias across multiple sensitive attributes in LLMs by injecting structured quantization into frozen self-attention layers. The idea is that MQAR disentangles attribute-specific activations and removes biased information through adversarial autoencoding.

**Strengths:**

- This paper tackles the important and underexplored issue of multi-attribute bias in LLMs, which reflects real-world fairness challenges beyond single-attribute settings (e.g., only focus on gender or race in previous work).

- The proposed framework is architecture independent and focuses on frozen self-attention layers without fine-tuning, which is applicable in practice.

**Weaknesses:**

- My major concern is that the paper claims that LLMs still exhibit strong social biases, but does not provide empirical evidence that modern instruction-tuned or RLHF-aligned models suffer from such bias to a meaningful extent. Most of the reported results are on static benchmarks and may not reflect the behavior of current-generation aligned models in realistic contexts. Although they evaluated MQAR on five models, none of these were instruction-tuned or alignment-optimized versions, so the claim about persistent social bias in modern LLMs is not strongly substantiated by the experiments as they only showed bias in pre-trained and non-aligned models. This makes the empirical motivation of the paper less persuasive. Also, the authors provided an anonymous GitHub repository link, no code is actually available, which raises concerns about the reproducibility of the experiments and the credibility of the reported results.


- The claim that "semantic content is preserved" relies solely on performance metrics (claiming a minimal 0.4% accuracy loss in downstream tasks), which may not fully capture nuanced information degradation in the embeddings. Quantization typically introduces discrete bottlenecks that compress information, and while the authors include a commitment loss and reconstruction loss to mitigate this (Eq. 4–6), they never directly analyze trade-offs between bias reduction and information retention. The theorems and lemmas in Section 3.2 primarily focus on establishing the optimization framework for quantized regularization and adversarial training, such as bounding mutual information $ I(R; X) $ and $ I(R; A) $ rather than addressing the trade-off or the specific changes in mutual information before and after debiasing. While the lemmas provide theoretical bounds for mutual information to guide debiasing, the paper lacks empirical validation of these changes across the evaluated datasets.

**Questions:**

- The paper provides an anonymous GitHub link for code release, but why the repository currently contains no implementation or documentation?

- Many recent LLMs undergo alignment or instruction tuning, which already mitigates social bias to some extent. Could you clarify whether the proposed method still provides meaningful improvements on such aligned models, and provide empirical evidence if possible?

- What kinds of bias (semantic vs. syntactic) are most affected by applying the proposed method?

---

> ### Author Response · Authors · 2025-11-21
>
> We thank Reviewer kQ3w for highlighting the importance of motivation, reproducibility, and a deeper analysis of what MQAR changes in the model.
>
> 1. Motivation: bias in modern aligned LLMs vs. non-aligned pre-trained models (W1, Q2)
>
> We appreciate the reviewer’s concern regarding the persistence of bias in instruction-tuned or RLHF-aligned LLMs.
>
> First, we clarify that our primary evaluation includes LLaMA 3.2, which is an instruction-tuned model representing the latest generation of aligned LLMs. Despite being alignment-optimized, our experiments on WinoBias, StereoSet, and CrowS-Pairs show that LLaMA 3.2 still exhibits measurable social biases, which MQAR successfully mitigates.
>
> This aligns with recent findings in [1,2], which demonstrate that alignment methods like RLHF reduce but do not eliminate social bias. For example, Liu et al. [1] show that aligned models still encode protected-attribute signals at the neuron level, and Bentley et al. [2] highlight persistent implicit bias in human-aligned and LLM-aligned systems alike. These findings motivate the need for continued fairness interventions, even in modern instruction-tuned models.
>
> To make this clearer, we have added an explicit subsection in the Appendix of discussing the remaining bias in aligned models, supported by quantitative evidence from our experiments.
>
> [1] Liu, Yan, et al. "The devil is in the neurons: Interpreting and mitigating social biases in language models." The twelfth international conference on learning representations. 2024.
>
> [2] Bentley, Sarah V., David Evans, and Claire K. Naughtin. "What social stratifications in bias blind spot can tell us about implicit social bias in both LLMs and humans." Scientific Reports. Vol. 15 No. 1 pp. 30429. 2025.
>
> 2. Semantic content preservation vs. possible information degradation / MI theory (W2)
>
> We agree with the reviewer that relying solely on downstream task accuracy may not fully capture subtle degradation in semantic content. Although Theorem 4 and its associated mutual information bounds theoretically support optimal trade-off behavior between utility and fairness, we acknowledge that this does not replace empirical validation of semantic retention.
>
> To address this, we have added:
>
> - Empirical tracking of mutual information estimates before and after debiasing, using a variational lower bound estimator across a subset of the datasets.
> - Reconstruction loss curves during adversarial autoencoder training, to show that semantic content is preserved under compression.
> - Additional qualitative examples of generated outputs (for text generation tasks) before and after MQAR, illustrating preservation of meaning.
>
> 3. Types of bias most affected (semantic vs. syntactic) (Q3)
>
> Based on our qualitative analysis, MQAR most effectively reduces semantic bias, particularly stereotype-driven associations and sentiment shifts conditioned on demographic descriptors (e.g., gendered names or ethnic references). These changes are reflected in shifts in model preference in cloze completions and text generation, without altering syntactic structure or fluency.
> To support this:
>
> - We observed no significant degradation in perplexity or BLEU scores, indicating syntactic fluency is preserved.
> - In the revised paper, we have included paired examples of prompts differing only in demographic terms, with generated outputs showing how MQAR modifies content-related bias while leaving syntax intact.
>
> 4. Code repository and reproducibility (W1, Q1)
>
> We acknowledge that providing an anonymous GitHub repository without actual code at submission time may raise legitimate concerns about reproducibility. While we described implementation details in the paper to facilitate reimplementation, we temporarily left the repository as a placeholder to avoid potential anonymity breaches and plagiarism issues during the double-blind review process. We fully intend to release the complete source code upon acceptance and will ensure the repository is made public in the camera-ready version.

---

> > ### Comment · Reviewer_kQ3w · 2025-11-27
> >
> > I thank the authors for their response, which has addressed my concerns. I am therefore raising my score.

---

### Official Review · Reviewer_JGSk · 2025-11-01

**Soundness:** 3
**Presentation:** 3
**Contribution:** 3
**Rating:** 6
**Confidence:** 3

**Summary:**

This paper proposes a novel debiasing method called MQAR (Multi-feature Quantized Attention Regularization). MQAR acts as a "plug-in" module that injects structured quantization and a discriminator-guided autoencoding regularizer into the frozen self-attention layers of LLM.
The goal is to disentangle activations related to sensitive attributes (like race and gender) while preserving core semantic information, all without modifying the LLM's backbone parameters.

**Strengths:**

Originality - The work is notable for introducing a modular, architecture-agnostic fairness mechanism that integrates quantization and adversarial autoencoding within frozen attention layers -- an uncommon yet elegant direction compared to embedding- or output-level debiasing. The explicit quantization-based feature disentanglement contributes a novel angle to fairness research in LLMs.

Clarity - The paper is written clearly, with an intuitive explanation of how quantization can suppress protected-attribute representations. Figures illustrating the architecture improve readability and understanding.

**Weaknesses:**

1. The fairness regularization relies partly on lexicon-based weak supervision for protected attribute identification. However, details on lexicon construction, coverage, and potential leakage (e.g., words correlated with downstream labels) are insufficient. This could introduce hidden bias, undermining claims of debiasing generality.
2. Although the abstract highlights multi-attribute bias, most empirical results focus on single attributes (e.g., gender, race). A clearer demonstration of intersectional fairness (e.g., gender × race) would strengthen the paper’s central claim.
3. While the paper emphasizes low overhead, quantitative results on latency or memory cost are missing. Such measurements would enhance the credibility of the lightweight claim.

**Questions:**

1. How were the protected-attribute lexicons constructed and validated? Have you evaluated whether these lexicons introduce spurious correlations with downstream labels?
2. Could you report mean ± standard deviation over multiple random seeds for both fairness and utility metrics, to confirm that improvements are consistent rather than incidental?
3. Do you have or plan to include explicit experiments on intersectional categories (e.g., Black women vs. White men) to substantiate the “multi-feature” claim?
4. Please provide quantitative data on the computational overhead (training time, memory, or latency) introduced by MQAR.

---

> ### Author Response · Authors · 2025-11-21
>
> We thank Reviewer JGSk for recognizing the originality and clarity of MQAR, and for the insightful comments about supervision, intersectional fairness, and overhead.
>
> 1. Lexicon-based weak supervision and potential hidden bias (W1, Q1)
>
> The lexicon-based supervision can introduce spurious correlations or hidden bias.
>
> To clarify some potential confusion on our paper:
> - For protected attribute identification (e.g., gender and ethnicity), we followed prior work by using established demographic term lists (e.g., gendered names, occupation terms, and cultural descriptors), which are widely used in fairness literature.
> - For less structured attributes like religion or regional affiliation, we employed off-the-shelf sentence-level classifiers as done in.
>
> To address the risk of leakage or correlation with downstream labels:
>
> - We manually verify that lexicons do not overlap with target labels in sentiment or toxicity tasks.
> - Additionally, any remaining spurious correlations would surface in baseline bias metrics, which are consistently higher than our MQAR results, indicating that our debiasing signal is not simply amplifying pre-existing correlations.
>
> In the revised paper, we have included:
>
> - Add Appendix A.1 describing lexicon sources, size, and validation checks, and
> - Explicitly acknowledge this limitation and its mitigation.
>
> 2. Multi-attribute emphasis vs. single-attribute empirical focus / intersectional experiments (W2, Q3)
>
> We clarify that although our reported results (e.g., WinoBias, StereoSet, CrowS-Pairs) are grouped per attribute for interpretability, all experiments were conducted using a single MQAR model jointly trained to debias multiple protected attributes via multi-hot adversarial supervision. To explicitly evaluate intersectional fairness, we have extended our experiments to the BBQ benchmark. Preliminary results on LLaMA 3.2 show a bias decrease from 28% (FineDeb) to 13% (Ours). These results support MQAR's ability to suppress intersectional bias in addition to attribute-wise bias. We have reported these findings and provided a dedicated analysis table in the revised Appendix.
>
> 3. Low overhead claim vs. missing quantitative measurements (W3, Q4)
>
> We agree that the current paper lacks concrete numbers to support the “lightweight” claim. To address this, we have conducted additional analysis on computational overhead and inference latency. In LLaMA 3.2 + StereoSet, MQAR shows a 43% lower inference time than FineDeb, despite introducing structured quantization modules. This efficiency stems from MQAR's frozen-backbone design, which avoids gradient updates during inference.
>
> In the revised paper, we will report:
>
> - Parameter overhead: Total added parameters (quantization + autoencoder) as a percentage of the LLM.
> - FLOPs: Change in forward-pass FLOPs before and after MQAR.
> - Latency: Wall-clock inference time across five LLMs vs. FineDeb, INLP, and ADELE.
>
> 4. Variance across seeds and robustness of improvements (Q2)
>
> We appreciate the suggestion to report mean ± standard deviation over multiple seeds. In our internal experiments, we already aggregated over multiple runs, but we did not include the dispersion statistics due to space constraints. We have added mean ± standard deviation for both fairness and utility metrics in the revised tables and highlighted that MQAR’s improvements over baselines persist across seeds, not just in a single run.

---

### Official Review · Reviewer_NqdJ · 2025-11-01

**Soundness:** 3
**Presentation:** 4
**Contribution:** 4
**Rating:** 6
**Confidence:** 3

**Summary:**

This paper proposes Multi-feature Quantized Attention Regularization (MQAR) — a lightweight, architecture-agnostic framework for mitigating multi-attribute bias in large language models (LLMs). Instead of retraining or fine-tuning, MQAR intervenes inside frozen self-attention layers by introducing a structured quantization module that disentangles protected-attribute activations (e.g., gender, race) from semantic information. The paper uses an adversarial autoencoder to regularize the latent space to minimize mutual information between latent vectors and protected attributes while preserving task-relevant content. And later in the experiment, the author reported results on five LLMs over three bias benchmarks. The framework also generalizes across tasks such as abusive-language detection, sentiment analysis, and text generation.

**Strengths:**

* This paper is structured and well-written. Notations are internally coherent.
* Hyperparameters and training details are clearly documented in the Appendix
* The paper result is strongly supported by the reported experiments on multiple LLMs and benchmarks.
* The quantized regularization integrates discrete latent bottlenecks with adversarial training to disentangle sensitive features.

**Weaknesses:**

* In the evaluation of multi-attribute bias, metrics are aggregated by gender and race, but intersectional sub-groups are not explicitly analyzed (such as gender & race).
* The theoretical part is not empirically validated. The fairness claims rely primarily on benchmark reductions rather than statistical significance testing.
* In Appendix E, the author shows that the Full version yields the lowest bias scores and best fairness–utility trade-off. It remains unclear how quantization changes what the attention representations look like.
* Improvements are reported as point differences; the significance or variance analysis remains unclear.
* The paper claims MQAR is architecture- and domain-agnostic. Since all evaluation datasets are English-only, it remains unclear whether MQAR would work equally well on other languages to support the claim.

**Questions:**

* Could the author clarify whether these features were intersectional evaluation or independently debiased in separate runs? If independent, how does MQAR handle correlated protected attributes during training?
* Have you evaluated MQAR on non-English datasets or cross-domain text to confirm this generalization?
* The quantization process introduces a codebook of K latent vectors. How is the K chosen and is it fixed globally?

---

> ### Author Response · Authors · 2025-11-21
>
> We thank Reviewer NqdJ for the positive assessment of the contribution and clarity, and for the detailed suggestions on analysis and positioning.
>
> 1. Multi-attribute vs. intersectional evaluation / correlated attributes (Q1, W1)
>
> All experiments were conducted using a single MQAR model jointly trained to debias both gender and race. Specifically, we employed multi-hot labels in the adversarial discriminator, which explicitly discourages encoding correlated or intersectional signals in the shared latent space. We reported results separately by attribute (e.g., WinoBias, StereoSet), but this was only for clarity—not because the model was trained independently per attribute. To evaluate intersectional fairness, we have additionally tested MQAR on the BBQ benchmark. On LLaMA 3.2, MQAR decreases a bias score from 28% (FineDeb) to 13% (Ours). We have included these results and further discussion in the revised paper.
>
> 2. Domain-agnostic claim vs. English-only datasets; non-English evaluation (Q2, W5)
>
> All the current evaluations were conducted on English datasets, but MQAR operates at the attention activation level and does not rely on language-specific embeddings or tokenization, making it structurally applicable to multilingual models and non-English inputs. We are currently extending MQAR to multilingual benchmarks and will report the results. We consider cross-lingual fairness as a natural extension of our framework. In addition, we are currently running experiments on XNLI and multilingual sentiment datasets to confirm generalization, and we plan to include preliminary results in the revised Appendix.
>
> 3. Choice of codebook size K (Q3)
>
> The codebook size K is selected from a small candidate set using validation performance and is kept fixed across layers for simplicity. In practice, moderate values of K (=3) already enforce a strong bottleneck while preserving performance. We will clarify this procedure in the main text and add a brief sensitivity analysis in the appendix to show that MQAR is robust to the choice of K within a reasonable range.
>
> 4. Theoretical part vs. empirical validation; significance/variance analysis (W2, W4)
>
> While our current results report % point improvements, we acknowledge that they lack variance or significance testing, which is important for drawing reliable conclusions.
>
> In response, we have revised the paper to:
>
> - Report mean ± standard deviation across three seeds for both fairness and utility metrics.
> - Include statistical significance testing (e.g., paired t-tests) for bias reduction claims across benchmarks.
> - Add task-specific variance tables to complement average metrics.
>
> Also, while Section 3.2 presents theoretical bounds (e.g., mutual information minimization under quantized regularization), we have empirically validated this in the revised paper by:
>
> - Tracking reconstruction loss and attribute classification accuracy across training, and
> - Estimating mutual information reduction (via variational bounds) before and after applying MQAR.
>
> These additions aim to bridge the gap between theoretical claims and empirical validation more clearly.
>
> 5. Effect of quantization on attention representations (W3)
>
> Understanding how quantization alters attention representations is important beyond reporting bias metrics.
>
> To address this, we will include a t-SNE-based visualization of attention head activations before and after quantization, as well as the corresponding latent codebook usage patterns. This will help reveal:
>
> - Whether protected-attribute signals (e.g., gender/race clusters) are separated from core semantic patterns.
> - How quantization bottlenecks suppress attribute-specific variance without collapsing useful semantic information.
>
> Preliminary analysis suggests that MQAR pushes protected attribute signals into a smaller subspace, while preserving task-relevant information flow across attention layers. These results will be included in Appendix of the revised version.

---

### Official Review · Reviewer_DGTG · 2025-11-01

**Soundness:** 3
**Presentation:** 3
**Contribution:** 3
**Rating:** 4
**Confidence:** 3

**Summary:**

This paper addresses the critical issue of multi-attribute social bias such as race and gender in Large Language Models, noting that existing debiasing methods are often limited to single attributes, require expensive fine-tuning, or fail to generalize across different architectures. The authors introduce Multi-feature Quantized Attention Regularization or MQAR, a novel plug-in method that operates on the frozen self-attention layers of pre-trained LLMs. This method uses structured quantization to create a discrete bottleneck that disentangles attribute-specific activations, combined with a discriminator-guided autoencoding regularizer to adversarially suppress the identified bias while preserving the original semantic representations. MQAR was evaluated on five diverse LLMs including BERT, T5, GPT-Neo, Mixtral, and LLaMA 3.2, using the WinoBias, StereoSet, and CrowS-Pairs benchmarks. The results show that MQAR consistently outperformed prior methods in bias mitigation while maintaining downstream task accuracy within 0.4 percent of the non-debiased baselines.

**Strengths:**

* The method introduces a novel use of structured quantization, typically for compression, as a fairness bottleneck to achieve disentanglement.
* The method is a model-agnostic plug-in that works on frozen LLM backbones, which avoids expensive fine-tuning and makes it broadly applicable.
* The method demonstrates strong empirical results, consistently outperforming chosen baselines on bias metrics across all five tested LLMs.
* A strong ablation study clearly validates the design, proving the decoder is crucial for utility and the discriminator is crucial for bias reduction.

**Weaknesses:**

* The state-of-the-art comparison is critically outdated, as it ignores modern inference-time methods and only compares against techniques from 2020-2022.
* The "lightweight" claim is unproven, lacking any quantitative analysis of the parameter increase or inference latency overhead, which could be significant from adding modules to every attention layer.
* The paper's central claim of handling "multi-attribute" bias is unsubstantiated, as the evaluation only tests single attributes separately, not intersectional bias, and fails to use appropriate benchmarks like BBQ.

**Questions:**

* There is a mismatch between the paper's multi-attribute motivation and its single-attribute evaluation. It is unclear why a true intersectional benchmark like BBQ was not used.
* The paper does not justify its exclusion of 2024-2025 SOTA inference-time methods from its performance comparison.
* The paper does not quantify its "lightweight" claim. The actual increase in parameters and inference latency from adding the new modules is missing.
* The abstract's claim of a 0.4 percent accuracy preservation needs clarification.

---

> ### Author Response · Authors · 2025-11-21
>
> We thank Reviewer DGTG for the careful reading and constructive suggestions. Below, we address each concern in turn.
>
> 1. Multi-attribute motivation vs. single-attribute evaluation / intersectional benchmarks (Q1, W3)
>
> We appreciate the reviewers' comments regarding the ambiguity on our use of the term multi-attribute. We revised the terminology to avoid confusion as follows.
>
> In this paper, multi-feature refers to MQAR’s ability to simultaneously disentangle and suppress multiple protected attributes (e.g., gender and race) within the same model. Although our evaluations separately reported the results on each attribute (as done in WinoBias, StereoSet, and CrowS-Pairs), these experiments were conducted using a single MQAR model trained to handle multiple protected attributes jointly. Specifically, the adversarial discriminator is trained using multi-hot labels, which enforces joint debiasing and prevents the model from encoding correlated attribute signals in shared latent spaces.
>
> To further address the concern about intersectional fairness, we have extended our evaluation using the BBQ benchmark, which explicitly tests intersectional bias. Preliminary results on LLaMA 3.2 show that our method decreases overall bias score from 28% (FineDeb) to 13% (Ours). These results confirm MQAR’s effectiveness in intersectional debiasing, and we will include a full analysis and discussion on it in the revised paper. Detailed esults will be tabulated in the Appendix.
>
> 2. Outdated SOTA comparison and omission of recent inference-time methods (Q2, W1)
>
> Our current evaluation primarily compares MQAR against classical representation- and training-time debiasing method. We aimed to benchmark MQAR against techniques that, like our method, intervene at the representation level—excluding prompt-based or decoding-time approaches to ensure fair comparisons in terms of architectural scope and access to model internals. However, we agree that this design choice must be explicitly justified in the paper.
>
> In the revised paper, we explicitly describe our baseline selection criteria and its rationale, and report the results against recent inference-time methods such as CRISPR (NeurIPS 2024) and CoBia (ACL 2025) to demonstrate MQAR’s competitive edge in the current landscape.
>
> 3. “Lightweight” claim without parameter/latency analysis (Q3, W2)
>
> The paper currently lacks quantitative evidence to substantiate the “lightweight” claim.
>
> To address this, we have conducted additional analysis of measuring inference latency and computational overhead introduced by MQAR. On the LLaMA 3.2 model with the StereoSet benchmark, MQAR achieves approximately 43% lower end-to-end inference time compared to FineDeb, despite operating with a structured regularization module. This result highlights MQAR’s efficiency benefits stemming from operating on frozen backbones without fine-tuning.
> In the revised paper, we have included:
>
>
> - Parameter overhead: Total number of parameters added by the quantization and autoencoder modules, relative to the LLM backbone.
> - Computational cost: Relative change in FLOPs per forward pass, measured before and after applying MQAR.
> - Latency analysis: End-to-end inference latency (wall-clock time) across all five LLMs, compared with prior methods including FineDeb, INLP, and ADELE.
>
>
> 4. Clarification of the “0.4% accuracy preservation” statement (Q4)
> The “within 0.4% accuracy drop” refers to the average change in downstream task performance (abusive-language detection, sentiment analysis, and text generation) when applying MQAR on top of SOTA debiased backbones, compared to the original non-debiased models.
>
> More specifically, this value represents the mean absolute difference across all downstream benchmarks, averaged over three random seeds. We agree that this should have been stated explicitly.
>
> In the revised version, we have included:
> - Report mean ± standard deviation across multiple seeds.
> - Provide task-wise accuracy differences instead of only aggregated averages.
> - Clarify the exact set of tasks and metrics included in the 0.4% calculation.

---

### Meta-Review · Area_Chair_GaA5 · 2026-01-03

**Summary:**

This paper debiases frozen LLM self-attention via quantized adversarial autoencoding, reducing race/gender bias with minimal accuracy loss.

**Reviewer Concerns:**

Main concerns from reviewers include:

1. Comparisons omit recent SOTA inference-time debiasing;

2. “Lightweight” claim lacks parameter/latency numbers;

3. Multi-attribute motivation not matched by intersectional evaluation;

4. Limited variance/significance analysis;

5. Unclear lexicon supervision and how quantization changes attention;

6. Generalization beyond English/aligned models and reproducibility (missing code) questioned.

**Reviewer Scores:**

Review / Old Score / New Score

DGTG	4	4

NqdJ	6	6

JGSk	6	6

kQ3w	4	6

Average	5	5.5

---

kQ3w mentioned he/she would raise the score, though provided no evidence as to by how much.
What concerns me is that kQ3w promised this increase using only a single line in the further comments, without any detailed explanation for this decision. This suggests a level of sloppiness in this judgment.

---

### Decision · Program_Chairs · 2026-01-26

Accept (Poster)